# B Cells and Double-Negative B Cells (CD27^−^IgD^−^) Are Related to Acute Pancreatitis Severity

**DOI:** 10.3390/diseases12010018

**Published:** 2024-01-05

**Authors:** Filipa Malheiro, Miguel Ângelo-Dias, Teresa Lopes, Sofia Azeredo-Lopes, Catarina Martins, Luis Miguel Borrego

**Affiliations:** 1Internal Medicine Department, LUZ SAÚDE, Hospital da Luz Lisboa, 1500-650 Lisboa, Portugal; 2CHRC, NOVA Medical School|Faculdade de Ciências Médicas (NMS|FCM), Universidade Nova de Lisboa, 1099-085 Lisboa, Portugal; miguel.dias@nms.unl.pt (M.Â.-D.); maria.lopes@nms.unl.pt (T.L.); sofia.azeredo@nms.unl.pt (S.A.-L.); catarina.martins@nms.unl.pt (C.M.);; 3Immunology Department, NOVA Medical School|Faculdade de Ciências Médicas (NMS|FCM), Universidade Nova de Lisboa, 1099-085 Lisboa, Portugal; 4Department of Statistics and Operational Research, Faculdade de Ciências, Universidade de Lisboa, 1749-016 Lisboa, Portugal; 5Immunoallergy Department, LUZ SAÚDE, Hospital da Luz Lisboa, 1500-650 Lisboa, Portugal

**Keywords:** acute pancreatitis, severity, lymphocyte, B cell, T cell, prognosis, biomarker

## Abstract

Acute pancreatitis (AP) is an increasingly frequent disease in which inflammation plays a crucial role. Fifty patients hospitalized with AP were included and peripheral blood samples were analyzed for B and T cell subpopulations at the time of hospitalization and 48 h after diagnosis. The Bedside Index of Severity in Acute Pancreatitis (BISAP) and length of hospital stay were also recorded. A healthy control (HC) group of 15 outpatients was included. AP patients showed higher neutrophil/lymphocyte (N/L) ratios and higher percentages of B cells than the HC group. The total B cell percentages were higher in patients with moderate/severe AP than in patients with mild AP. The percentages of B cells as well as the percentages of the CD27^−^IgD^−^ B cell subset decreased from admission to 48 h after admission. The patients with higher BISAP scores showed lower percentages of peripheral lymphocytes but higher percentages of CD27^−^IgD^−^ B cells. Higher BISAP scores, N/L ratios, and peripheral blood B cell levels emerged as predictors of hospital stay length in AP patients. Our findings underscore the importance of early markers for disease severity. Additionally, the N/L ratio along with the BISAP score and circulating B cell levels form a robust predictive model for hospital stay duration of AP patients.

## 1. Introduction

Acute pancreatitis (AP) is an acute inflammatory disorder of the pancreas associated with significant morbidity and high health costs [1]. Hospitalizations due to AP have been increasing in recent decades [2]. The most common causes of AP are pancreatic ductal obstruction by gallstones, alcohol, and endoscopic retrograde cholangiopancreatography, which activate pathological cellular pathways and organelle dysfunction, leading to acinar cell death and subsequent local and systemic inflammation. The crosstalk between acinar cells and the immune system perpetuates inflammatory responses both locally and systemically [3]. Over the last few years, immune system activation has been identified as a key trigger and regulator of inflammatory injury in the pancreas, influencing the extent of pancreatic necrosis, systemic organ failure, and disease deterioration [4]. As an inflammatory process, acute pancreatitis results in excessive innate immune system activation and consequent release of pro-inflammatory mediators such as IL-6, IL-8, and TNF-α [5]. While the involvement of lymphocytes in this process has been partially explored through previous studies, the specific impact of alterations in peripheral lymphocyte subsets on prognostic accuracy, disease management, and hospitalization duration for patients with AP remains unclear [6,7]. Even though acute pancreatitis is a disease with high costs for the health care system, few studies have tried to relate the prediction of severity and length of hospital stay. Nevertheless, there have been some reports relating the length of hospital stay in patients with acute pancreatitis and the absolute lymphocyte counts observed [8].

Currently, there is an open quest to identify a biochemical marker that can be easily measured to assess and predict the severity of patients admitted with acute pancreatitis, and to monitor their clinical status during disease progression [9]. In fact, the current methods used in clinical practice involve many variables and some are not readily available to support decision making [10]. The identification of such biomarkers would enhance diagnostic accuracy, enable timely decision making, optimize resource planning, and facilitate the appropriate interventions for optimal patient outcomes. B cells have already been studied in patients with AP and the B regulatory subset of B cells has been shown to be useful in the prediction of the severity of AP [6,11]. Double-negative B cells (CD27^−^IgD^−^ B cells) are a rare B cell subset that constitutes about 5% of all peripheral B cells in healthy individuals [12]. This B cell subset has been poorly characterized for a long time until recent studies indicated their potential roles in diseases especially autoimmune diseases, some infections, and chronic inflammatory diseases [13]. As an inflammatory disease of the pancreas, it would be of interest to study this B cell subset in AP patients.

Understanding how changes in peripheral lymphocyte subsets relate to these aspects can provide valuable insights. Thus, in this study, we aimed to explore the peripheral immune cell subsets in acute pancreatitis patients and assess their association with disease severity. In brief, our goal was to evaluate whether immune cells can be used to determine the severity and length of hospital stay early in the course of the disease and therefore be used as biomarkers of acute pancreatitis severity.

## 2. Materials and Methods

### 2.1. Subjects and Sample Collection

In this prospective observational study, fifty (n = 50) AP patients admitted to the Hospital da Luz Lisboa between February 2021 and March 2023 were consecutively recruited. AP patients were diagnosed according to the revised Atlanta classification system which requires at least 2 of the following 3 features: abdominal pain characteristic of AP, serum amylase and/or lipase at least three times greater than the reference limit, and findings characteristic of acute pancreatitis on abdominal computerized tomography scan (CT scan) or transabdominal ultrasonography [14]. In all patients, the onset of abdominal pain and hospital admission was 48 h or less. The exclusion criteria eliminated patients with a previous hospitalization for AP in the last 6 months, pregnant women, patients with important uncontrolled comorbidities such as organ disease (cardiac, renal, hepatic) or terminal neoplasms, patients on immunosuppressive or chemotherapy, and patients younger than 18 years or older than 85 years old.

Patients were also classified according to the cause and severity, as defined by the revised Atlanta classification system [14]. The Bedside Index of Severity in Acute Pancreatitis (BISAP) score was also applied to all patients [15]. Patients were further classified into BISAP^low^ if they scored 0 or 1 and BISAP^hi^ if they scored ≥2.

Fifteen (n = 15) age and sex-matched healthy individuals were included as the healthy control (HC) group. The HCs were ambulatory individuals observed at the Hospital da Luz Lisboa, without previous pancreatic pathology or acute systemic disease.

All clinical data were analyzed, which included demographic characteristics: gender, age, body mass index (BMI), and comorbidities. AP clinical characteristics and outcomes such as cause and severity of AP, BISAP score, mechanical ventilation, ICU and hospital length of stay, as well as complications during hospitalization and mortality were also recorded. Additionally, data on other markers were collected from the hospital patient file, including C- reactive protein (CRP) levels and complete blood counts (CBCs) with platelets. Blood samples were collected from patients with acute pancreatitis in the first 24 h of diagnosis (T1) and at 48 h (T2) after hospitalization. The HC group had only one blood collection time point performed after recruitment. All samples were collected into EDTA tubes and analyzed within 24 h after collection.

This study was approved by the Hospital da Luz (CES/24/2020/ME) and NOVA Medical School (14/2019/ADENDA/CEFCM) ethics committees. Written informed consent was obtained from each subject before sample collection. This study was performed in accordance with the Declaration of Helsinki. All radiological, laboratory, and clinical data were anonymized before being analyzed.

### 2.2. Flow Cytometry

Aiming for standardized approaches that could be easily implemented in the routine practices of clinical laboratories, in this work, we used pre-validated flow cytometry panels for cell characterization. For B-cells, we used the DryFlowEx ASC screen Kit from Exbio (Praha, Czech Republic), a multicolor panel of antibody conjugates dried in a single flow cytometry tube, which includes CD45 Pacific Blue™, IgD FITC, CD27 PE, CD24 PerCP-Cy™5.5, CD19 PE-Cy™7, CD21 APC, and CD38 APC-Cy™7. The characterization of T cell subsets was performed according to the panel proposed by the Human Immunophenotyping Consortium, which includes CD25 PE, CD4 PerCP-Cy™5.5, CD127 APC, CD45RO APC-Cy™7, CCR4 PE-Cy-Cy™7, CD3 BV421™, and HLA-DR BV510™ [16]. All antibodies in this panel were from Biolegend (San Diego, CA, USA).

Briefly, for both panels, 100 µL of pre-washed whole blood cells were incubated with the monoclonal antibodies for 20 min at room temperature, lysed, washed, and acquired. To prevent non-specific polymer interactions, 50 µL of BD Horizon Brilliant Staining Buffer (BD Biosciences, San Diego, CA, USA) was also added.

All acquisitions were performed using an 8-color BD FACS Canto II Flow Cytometer (BD, San Jose, CA, USA) and BD FACS Diva software version 8.0.2. Data analyses were performed with Infinicyt^TM^ 2.0 (Cytognos, SL. Salamanca, Spain) and FlowJo^TM^ v10.6.2 (BD Life Sciences) software [16].

Detailed information on all antibody panels and gating strategies are presented in Appendix A and Appendix A, respectively.

### 2.3. Statistical Analysis

Categorical variables are presented as absolute frequencies and percentages and the associations between them were analyzed with Fisher’s exact test. For continuous normal distributed variables, means and standard deviations (SDs) are presented; for non-normal distributed data, the medians and interquartile ranges (IQR) were obtained. Comparisons between two independent samples were performed using the *t*-test and the *t*-test with Welch’s correction (as appropriate) or the nonparametric Mann–Whitney U-test. Paired group comparisons were performed using paired *t*-tests or the nonparametric Wilcoxon matched-pairs signed-rank test, as applicable. The Pearson test was used to assess correlations between continuous variables. For all tests, a *p* value < 0.05 was considered significant and are indicated as follows: * *p* < 0.05, ** *p* < 0.01, *** *p* < 0.001, **** *p* < 0.0001.

Descriptive analyses were performed with GraphPad Prism 9 software v9.0.2 (Graph Pad, San Diego, CA, USA).

Poisson regression models were used to assess the association between the number of days spent at the hospital and several other factors. A univariable analysis was performed to identify which variables could potentially explain the outcome variable—length of stay in hospital. Only those with a *p* value less than 0.25 were selected for the multivariable study. A stepwise method was used for variable selection, and the deviance residuals obtained by the models were used for testing the goodness-of-fit and model fit. Tests of equidispersion were performed to evaluate overdispersion in the data. The R Statistical software (v.4.2.3; R Core Team 2023) was used for fitting the Poisson regression models and its AER package to test for overdispersion [17].

## 3. Results

The baseline characteristics of the patients with acute pancreatitis and healthy controls are summarized in Table 1. Most of the cases were caused by gallstones and the mean length of stay was 6 days. Sixteen patients (32%) scored a BISAP of 2 or 3.

### 3.1. Immune Profile of Acute Pancreatitis Patients at Hospital Admission

As a first step to elucidate the features of immune responses during AP, we determined the blood levels of B and T cell subsets at hospital admission and compared them with those of the HC group. The patients with AP presented, at admission, with higher leucocyte counts (*p* < 0.0001) and altered CBC differentials. In AP patients, these differences were lower percentages of lymphocytes at admission (*p* < 0.0001), as well as a higher neutrophil/lymphocyte ratio (N/L ratio) (*p* < 0.0001) compared to the levels observed in the HCs.

Within lymphocytes, there were differences mainly in CD4^+^ T cells and B cells. Again, at T1, the AP patients showed decreased percentages of CD4^+^ T cells compared to the HCs (*p* = 0.037), but increased percentages of B cells (*p* = 0.007). Moreover, the circulating CD4^+^ T cell compartment was enriched in activated HLA DR^+^ cells in the AP patients at admission compared to the controls (*p* = 0.023), with a trend for lower naïve and higher memory CD4^+^ T cell percentages in patients (*p* = 0.080).

Regarding the other cell populations studied, including the different subsets of B cells, no further differences were considered significant between the AP and HC groups at T1.

The detailed information on the B and T cell subsets is depicted in Appendix A.

### 3.2. Dynamics of the Immune Profile in Patients with Acute Pancreatitis during the First 48 H after Diagnosis

As the first 48 h of hospitalization are crucial in the evolution of acute pancreatitis and this is the time when most severity scores are assessed, we decided to evaluate the dynamics of the immune profile during this period (Figure 1).

The AP patients had higher leucocytes counts at admission (*p* < 0.001) but higher CRP levels at T2 (*p* = 0.001). The N/L ratio was higher at T1 than at T2 (*p* < 0.001) and the lymphocyte percentages, within leucocytes, were lower at T1 than at T2 (*p* < 0.001). This information is summarized in Table 2. In the lymphocyte compartment, the T cell percentages were higher at T2 than at T1 (*p* = 0.001) and this was also true for CD4^+^ T cell percentages, which were higher after 48 h (*p* < 0.001), but there was a decrease in the percentages of activated HLA-DR ^+^ CD4^+^ T cells from T1 to T2 (*p* = 0.001). Finally, the T/B cell ratios were also lower at admission than at T2 (*p* = 0.001).

Within the circulating B cell compartment, there seemed to be a modification during the first 48 h of disease in the patients with AP. Thus, by assessing the early dynamics of B cells in the AP patients, we were able to identify that the total percentages of B cells (within lymphocytes) decreased from T1 to T2 (*p* = 0.004). Moreover, differentiated subsets such as double negative CD27^−^IgD^−^ B cells and activated CD21low CD38- B cells decreased from admission to 48 h after diagnosis (*p* = 0.020 and *p* < 0.001, respectively). This information is summarized in Appendix A.

### 3.3. Early Variations in the Immune Profile and Its Relation to Severity in Patients with Acute Pancreatitis

Recognizing that the clinical heterogeneity of the AP group could have an impact on the immune profiles observed, we further divided the patients according to their clinical features, and specifically according to their BISAP scores (Figure 2 and Table 3). In fact, in the subgroup of patients with higher BISAP scores (BISAP^hi^) who are expected to have increased disease severity, we observed higher CRP levels at both T1 and T2 (*p* = 0.024 and *p* = 0.010, respectively), and extended hospital stays compared to patients with lower BISAP scores (BISAP^low^, *p* = 0.035). Similarly higher neutrophil counts and percentages (Figure 2A,B) were observed in BISAP^hi^ patients compared to the BISAP^low^ (*p* = 0.009 and *p* = 0.020) and HC groups (*p* < 0.001). Lower lymphocyte percentages were also observed in BISAP^hi^ patients (*p* ≤ 0.025), though there was no difference in lymphocyte cell counts (Figure 2C,D). Likewise, the N/L ratio was higher in BISAP^hi^ patients (Figure 2E) compared to the BISAP^low^ (*p* = 0.040) and HC groups (*p* < 0.001). Moreover, dividing the patients according to disease severity showed that CD21^low^CD38^−^ B cells (*p* = 0.017) and IgD^−^CD27^−^ B cells (*p* = 0.015) were elevated in BISAP^hi^ patients compared to BISAP^low^ patients, though their levels were not statistically different from those of the HC group (Figure 2F,G). Additionally, the percentages of activated HLA-DR^+^ CD4^+^ T cells were higher in BISAP^hi^ patients (Figure 2H) compared to both the BISAP^low^ (*p* = 0.048) and HC (*p* = 0.015) groups (Figure 2H).

Using a parallel classification system for severity according to the revised Atlanta classification system, moderate/severe AP patients also showed increased serum CRP levels and leucocyte counts (Figure 3A). In addition, the frequencies of CD4^+^ T cells were lower in this group compared to the HCs (*p* = 0.047), but they were not different from those of mild AP patients (Figure 3B). However, the proportions of memory CD4^+^ T cells (Figure 3C) were higher in moderate/severe AP patients than in the mild AP (*p* = 0.022) and HC groups (*p* = 0.008). The total B cell percentages were also higher in patients with moderate/severe AP than in the HC group (*p* = 0.025). The detailed information is given in Table 4.

### 3.4. Length of Stay and the Immune Profile: Exploring Relations for Patient Monitoring and Follow Up

Recognizing the differences in the immune profile of the AP patients, we then aimed to identify their predictive potential in terms of length of hospital stay in these patients.

The univariable Poisson regression results for the candidate variables for the multivariable analysis are shown in Appendix A. All the remaining variables considered in this study obtained *p*-values > 0.25. The final multivariable Poisson models to assess the length of stay at hospital are described in Table 5. The multivariable Poisson model 1 was based on N/L ratio, % of B cells, and BISAP score values for the AP patients. In detail, after adjusting for the other variables, for a unit increase in the N/L ratio, there was an estimated 2.4% increase in the expected number of days in hospital (*p* < 0.001; 95% CI 1.015–1.033), while for B cells, there was an estimated 1.8% increase (*p* = 0.047; 95% CI 1.000–1.034) (Figure 4A). In addition, this model predicted that patients with a BISAP score of 2 or 3 will spend, on average, 34.3% more days at hospital (*p* = 0.031; 95% CI 1.024–1.755) when compared to patients with a BISAP score of 0 or 1 (Figure 4B).

## 4. Discussion

Acute pancreatitis is an inflammatory disease of the pancreas with high costs and is one of the main gastrointestinal causes for hospitalization in the developed world [18]. Predicting the severity of this disease as well as understanding its physiopathology is needed to reduce morbidity and mortality in patients with acute pancreatitis. Several risk scores, individual biomarkers, and radiological scoring systems have been developed to predict outcomes; these include the revised Atlanta classification system from 2012, the APACHE II score, serum CRP, the BISAP score, and Ranson’s criteria [14,19]. The main problem with these scores is that they are applied only 24–48 h after hospitalization and therefore, are not readily available to predict the severity of acute pancreatitis [20].

In our study, we described alterations in the immune profile of AP patients within the first 48 h after diagnosis, particularly in the circulating B cell compartment. Moreover, to the best of our knowledge, we are the first to identify potential immune population biomarkers for hospital stay length in AP patients, offering valuable insights into risk assessments of patients in the initial clinical approach. Indeed, in our cohort, higher BISAP scores, N/L ratios, and peripheral blood B cell frequencies emerged as robust predictors of hospital stay length in AP patients, which could have potential clinical applications.

It is now known that injured acinar cells of the pancreas release chemokines that lead to the infiltration of immune cells, firstly neutrophils, worsening the tissue injury of the pancreas and later leading to systemic inflammation [21,22]. Neutrophils activate trypsinogen in acinar cells. These cells amplify the inflammatory cascade, generating many chemokines and cytokines including IL-1, IL-6, and intercellular adhesion molecule 1 (ICAM-1) to promote pancreatic and extra pancreatic multiorgan injury [18]. In acute pancreatitis, after the activation of the innate immune system, a cascade of inflammation follows including the activation of the adaptative immune system. Disease severity depends on whether the sterile inflammatory response resolves or amplifies [23]. As compensatory anti-inflammatory responses occur, as shown by increases in regulatory T cells in lymphoid tissues, dysregulation, instead of balance restoration, and persistent inflammation or immunosuppression compensatory mechanisms may prevail [24].

Lymphopenia has already been shown to be related to systemic inflammatory responses and sepsis and several theories have been presented to explain these changes in the immune profile [25]. These theories include patients with a limited initial inflammatory phase which is followed by an immunosuppression pattern while other patients develop a state of prolonged immunosuppression that exposes them to secondary infections or to a recurrence of the initial unresolved infection [26].

In line with previous works, we observed that patients with acute pancreatitis have lower counts of lymphocytes at admission. In fact, lymphopenia is usually related to AP severity [3,27,28]. Similar to what happens in SIRS, peripheral blood lymphocyte depletion in acute pancreatitis may result from both excessive apoptosis and migration to the site of inflammation as has been previously hypothesized [29,30]. Regarding more detailed lymphocyte profiling, most studies have explored T lymphocytes, showing that acute pancreatitis results in the systemic activation of T cells [31,32]. Among T cells, it has been shown that there is a significant depletion of the CD4^+^ population, while CD8^+^ cell levels were reported to be present in the normal ranges [29]. We propose that the decrease in activated CD4^+^ T cell population is related to their migration to the site of inflammation.

In line with previous data, we also found that patients with acute pancreatitis have higher circulating B cell levels at admission, and that B cells will start to decrease within the following days as the disease progresses [11]. According to one investigation by Shi and collaborators, the levels of B cells at hospital admission of patients with acute pancreatitis may be of value for predicting the development of organ failure, even though this study did not include a control group and peripheral blood B cell levels were only measured once at the time of diagnosis of acute pancreatitis [33].

The inflammatory background of pancreatitis may be related to these observations. In fact, it has been suggested that despite inflammation being able to modulate hematopoiesis in the bone marrow (BM), promoting granulocytic pathways to the detriment of other lineages (i.e., B cells), growing evidence supports that inflammation mobilizes developing B cells from the BM to the periphery and even induces extramedullary lymphopoiesis [34]. Such mechanisms could explain the observations of increased proportions of circulating B cells in AP patients at hospital admission.

Moreover, B cells are expected to play a role in acute pancreatitis as it is an inflammatory disease to which they can contribute through the increased production of antibodies, antigen presentation, as well as inhibition of the activation and proliferation of other inflammatory cells by secreting anti-inflammatory factors. In fact, in terms of B cell subtypes, other authors have found that an increase in B regulatory cell levels occurs in mild cases of acute pancreatitis during the first week while the levels of these cells remain stable in patients with severe acute pancreatitis [6]. Still, the increase in B cell levels has been described in the normal response to inflammation and is probably related to the enhancement of resistance to infection but the exact role of increased B cell levels in peripheral blood of patients with AP is yet unknown, and neither subset of B cells is related to the evolution or severity of acute pancreatitis [33,35,36].

Moreover, the possible extramedullary B cell lymphopoiesis may occur in a “selection-light” environment, allowing for the survival of self-reactive B cells that would normally be deleted in the BM, supporting the recognized linkage between inflammation and the induction of systemic autoimmune disease [34].

Interestingly, in our study, despite no differences being found between the AP patients and controls in the other B cell subsets analyzed, there was a decrease in the percentages of circulating CD27^−^IgD^−^ B cells from admission (T1) to the second time point (T2) in the patients with AP. It is known that these cells have several similarities with conventional memory B cells. They are usually found in tonsils and in low numbers in healthy individuals but are expanded in several diseases, mainly autoimmune diseases [12]. They have been mostly studied in systematic lupus erythematosus, COVID-19, and malaria but also in low-grade chronic inflammatory disorders and some cancers [13]. CD27^−^IgD^−^ B cells can be further subdivided in four subtypes capable with varying functions in different conditions [13].

Moreover, in our study, we found that CD27^−^IgD^−^ B cells were related to disease severity when considering the BISAP score. Interestingly, we found that patients with a higher BISAP score had higher peripheral blood values of CD27^−^IgD^−^ B cells in a similar way as most chronic inflammatory diseases in which these cells are expanded. We also found that CD27^−^IgD^−^ B cells decreased from admission to 48 h after the diagnosis of acute pancreatitis. It is possible that the differentiation of this subset of B cells might occur as a response to inflammation and after differentiation, these cells migrate to the affected organ (pancreas), as has been described in some lung cancers [37]. Gong and collaborators have speculated, based on their study on the microenvironment of patients with nasopharyngeal carcinoma, that CD27^−^IgD^−^ B cells play a local immunosuppressive role in these tumors as an increase in this B cell subset is significantly correlated with disease progression in nasopharyngeal carcinomas [38]. The previous findings regarding human CD27^−^IgD^−^ B cells suggest that they may represent a heterogeneous B cell population with varying roles according to disease and we speculate that these cells are an early marker at hospital admission for the severity of AP. To the best of our knowledge, this is the first study to point out the changes in CD27^−^IgD^−^ B cells in patients with acute pancreatitis as well as relating higher levels of this B cell subset with higher BISAP scores.

The reason why this subset of B cells is expanded in patients with acute pancreatitis might provide insights into acute pancreatitis pathogenesis and help to develop future treatments. We also expect CD27^−^IgD^−^ B cells to rise in the peripheral blood of patients with long-term complications of acute pancreatitis such as infected necrosis and prolonged organ failure. For these reasons, it might be of interest to monitor them for an extended period of time beyond the first 48 h after the diagnosis of acute pancreatitis as was performed in our study. As CD27^−^IgD^−^ B cells may be involved in various diseases, including chronic inflammatory and infectious diseases, autoimmune diseases, as well as some neoplasms, research on this subset of B cells might be essential in understanding these conditions and their treatments. As an example, therapies targeting CD27^−^IgD^−^ B cells may improve clinical outcomes in chronic conditions, while understanding the reasons why this subset of B cells is increased locally in some cancers may provide insights into the progression of these neoplasms to eventually develop new immunotherapeutic strategies.

Acute pancreatitis is a high-cost disease with a median cost of nearly USD 7000 per hospitalization in 2013 [39]. The hospital length of stay, costs, and mortality attributed to acute pancreatitis have declined in 2009–2013 compared with 2002–2005 [40]. These improved outcomes have been attributed to several factors, including increased efficacy of diagnosis, routine use of risk stratification tools, and more aggressive management in intensive care units for the most severe cases [41]. However, few studies have determined which factors might relate to a longer length of stay of patients with acute pancreatitis except for those with a higher severity score using the revised Atlanta classification and some biomarkers such as the C reactive protein/albumin ratio [42,43].

In our study, the N/L ratio was higher in patients with higher BISAP scores. The N/L ratio has been shown to be an independent predictor of prognosis in various clinical situations, including sepsis, some malignancies, and cardiovascular disease [44]. The N/L ratio is a biomarker that joins the innate immune system due to neutrophils and the adaptive immunity supported by lymphocytes. It has been used as an independent prognostic factor for morbidity and mortality in several diseases including stroke, myocardial infarction, infection, and diseases that activate SIRS [45].

In our model of linear regression, in this cohort of patients, a higher BISAP score, N/L ratio, and frequency of peripheral blood B cells were the best predictors of the length of stay of patients with AP. The N/L ratio has been already described as statistically significant predictor of the severity of acute pancreatitis upon presentation of the patient to the hospital [46]. The BISAP score has been largely used because of the simplicity of its calculation besides being accurate for risk stratification of patients with acute pancreatitis [15]. To our knowledge, this is the first study to consider the BISAP score, peripheral blood B cell levels, and N/L ratio as predictors of length of hospital stay of AP patients. Nevertheless, our model 2 is also interesting to consider as it includes the BISAP score, cause of AP, % of B cells, and HLA-DR^+^ CD4^+^ T cells, which can be easily measured in clinical research laboratories.

We consider the main limitations of our study to be the small number of patients with severe AP as most patients had a BISAP score <2 or had mild or moderate AP according to the revised Atlanta classification system. An increased number of HCs would also be of interest. In addition, we acknowledge that conducting a more extended follow-up and a later determination of immune profile dynamics, along with a comprehensive investigation of the various lymphocyte subtypes involved in the severity of AP, could be valuable for gaining insights into the pathophysiology of AP. Unlike neoplastic diseases, the acquisition of tissue samples from patients with acute pancreatitis is difficult and not feasible, which limits the scientific research to some extent. Therefore, the evidence of immunological alternations in patients with acute pancreatitis has been obtained mainly by studying peripheral blood. Peripheral blood, besides being accessible to study, also reflects the systemic component of acute pancreatitis as previously described.

We believe there is much to learn about the pathogenesis of acute pancreatitis and the role of the adaptative immune system in acute pancreatitis. The exact role of B cells and T cells and their subsets still need to be elucidated. An early single marker of disease severity is much needed for clinical practice. Moreover, future studies need to determine the dynamic evolution of the levels of these cells in peripheral blood during the course of acute pancreatitis. Are the changes in the levels in peripheral blood a cause or consequence of higher levels of inflammation and SIRS? And can we use the variations in the levels of these cells to our advantage and point out a useful biomarker of acute pancreatitis severity? This is especially important in order to develop future treatments that can balance the excess inflammation and the risk of late infections in patients with acute pancreatitis.

## 5. Conclusions

The findings on CD27^−^IgD^−^ B cells in AP patients, especially among those with higher BISAP scores, underscore the need for further investigations into the evolving patterns of these cells during the disease course. Such inquiries hold the potential to yield valuable biomarkers and therapeutic approaches for the improved management of patients with acute pancreatitis Additionally, our study underscores the significance of the BISAP score, N/L ratio, and peripheral blood B cell frequency as predictive tools for assessing the hospital length of stay of AP patients, providing fresh insights into patient risk assessment and care.

## Figures and Tables

**Figure 1 diseases-12-00018-f001:**
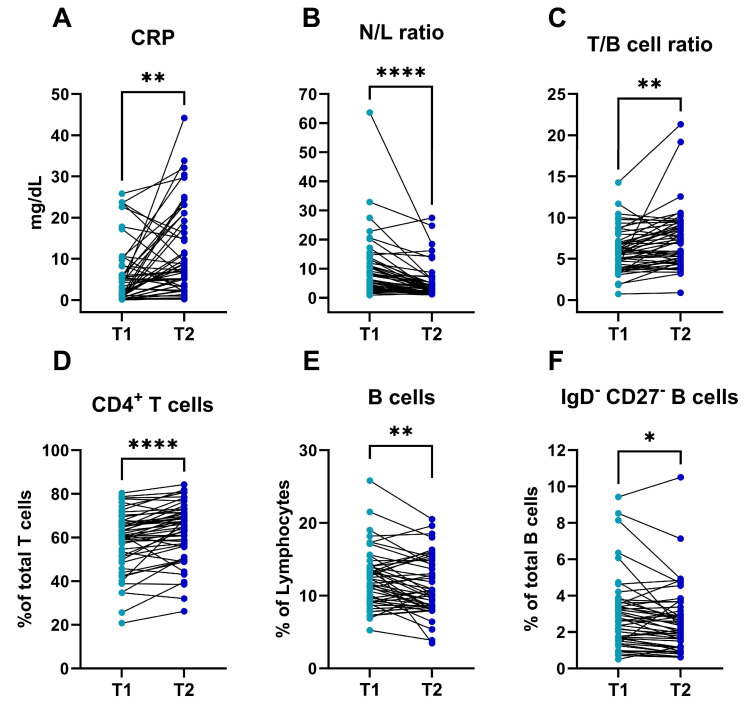
Inflammatory and cellular immune dynamics in AP patients during the first 48 h after diagnosis. Concentration of CRP (mg/dL) (**A**); N/L ratio (**B**); T/B cell ratio (**C**); (**D**–**F**) percentage of CD4^+^ T cells (**D**), B cells (**E**), and DN B cells (**F**) in AP patients (n = 50) at admission (T1) and 48 h after diagnosis (T2). Significance determined by paired *t* tests (**A**,**C**–**F**) and Wilcoxon matched-pairs signed-rank test (**B**). AP, acute pancreatitis; HC, healthy control; CRP, C reactive protein; N/L, neutrophil/lymphocyte. * *p* < 0.05; ** *p* < 0.01; **** *p* < 0.0001.

**Figure 2 diseases-12-00018-f002:**
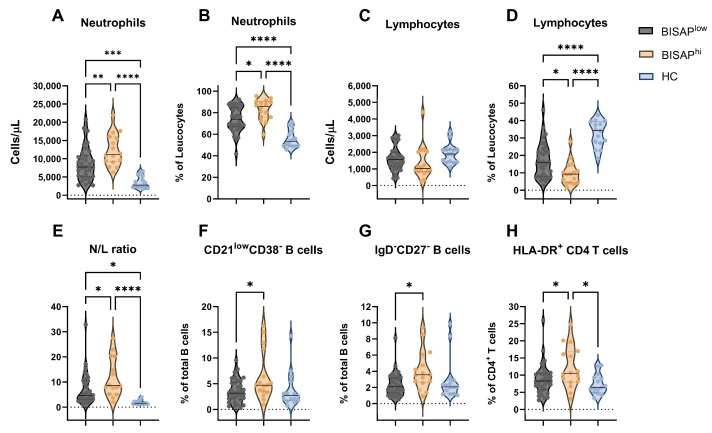
Comparison of cellular immune distributions between BISAP^hi^, BISAP^low^, and HC groups at admission. Concentration and frequency of circulating neutrophils (**A**,**B**) and lymphocytes (**C**,**D**). (**E**) N/L ratio. (**F**–**H**) frequency of CD21^low^CD38^−^ B cells (**F**), IgD^−^CD27^−^ B cells (**G**), and HLA-DR^+^ CD4^+^ T cells (**H**). Violin plots represent median ± IQR. Significance determined by one-way ANOVA. * *p* < 0.05; ** *p* < 0.01; *** *p* < 0.001; **** *p* < 0.0001. N/L, neutrophil/lymphocyte.

**Figure 3 diseases-12-00018-f003:**
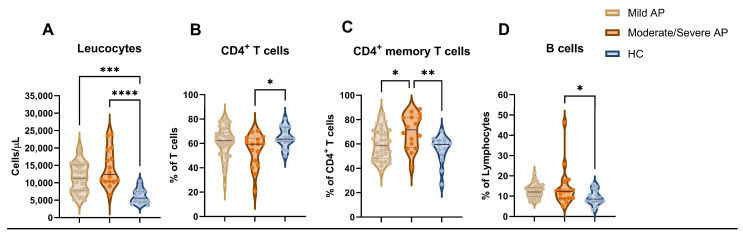
Comparison of cellular immune distributions between mild AP, moderate/severe AP, and HC groups at admission. Concentration of circulating leucocytes (**A**) and frequency of circulating CD4^+^ T cells (**B**), CD4^+^ memory T cells (**C**), and B cells (**D**). Violin plots represent median ± IQR. Significance determined by one-way ANOVA. * *p* < 0.05; ** *p* < 0.01; *** *p* < 0.001; **** *p* < 0.0001. AP, acute pancreatitis; HC, healthy control.

**Figure 4 diseases-12-00018-f004:**
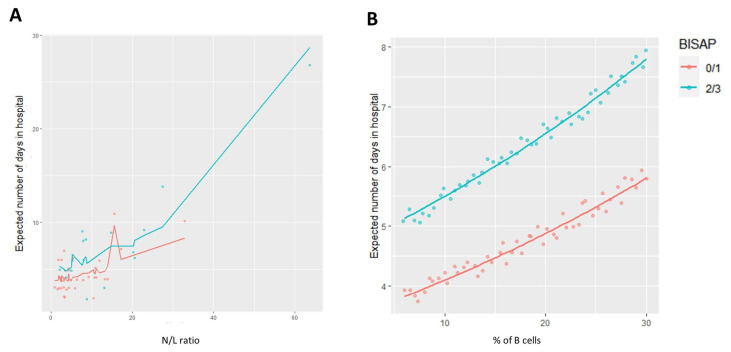
Prediction of the expected number of days of hospitalization in the analyzed cohort of patients with low and high BISAP scores based on (**A**) N/L ratio and (**B**) % of B cells.

**Table 1 diseases-12-00018-t001:** Demographic characteristics of patients with AP and HCs.

Characteristics	AP (n = 50)	HC (n = 15)	*p*-Value
Age, years, mean (SD)	59.9 (14.6)	61.1 (14.3)	n.s. ^a^
Gender, n (%)			n.s. ^b^
Male	23 (46)	7 (47)	
Female	27 (54)	8 (53)	
BMI, kg/m^2^, mean (SD)	29.0 (5.8)	26.8 (3.1)	n.s. ^a^
Cause (%)		-	-
Gallstone	23 (46)		
Alcoholic	5 (10)		
Unknown	21 (42)		
Other	1 (2)		
Severity, n (%)		-	-
Mild	34 (66)		
Moderate	13 (28)		
Severe	3 (6)		
30-day mortality, n (%)	0 (0)	-	-
ICU, n (%)	3 (6)	-	-
Mechanical ventilation, n (%)	1 (2)	-	-
BISAP score, n (%)		-	-
0	14 (28)		
1	20 (40)		
2	9 (18)		
3	7 (14)		
AP criteria, n (%)		-	-
2	26 (52)		
3	24 (48)		
LOS, days, median [IQR]	6 [3–6]	-	-

Abbreviations: AP, acute pancreatitis; HC, healthy control; BMI, body mass index; ICU, intensive care unit; LOS, length of stay at hospital; IQR, interquartile range; SD, standard deviation; n.s., not significant. ^a^ Unpaired *t* test with Welch’s correction; ^b^ Fisher’s exact test.

**Table 2 diseases-12-00018-t002:** Complete blood counts, neutrophil-to-lymphocyte ratios, and CRP levels in AP patients and HCs.

	AP at Admission (T1) (n = 50)	AP at 48 h (T2) (n = 50)	HC (n = 15)	T1 vs. HC ^a^	T2-T1 ^d^
**Leucocytes, cells/µL**	12,395 (4733)	8785 (4257)	6197 (2077)	**<0.001**	**<0.001**
** Neutrophils, %**	76.76 (11.89)	65.98 (12.85)	55.76 (8.23)	**<0.001 ^b^**	**<0.001**
** Neutrophils, cells/µL**	9864 (4764)	6160 (4148)	3570 (1642)	**<0.001**	**<0.001**
** Eosinophils, %**	1.10 (1.26)	2.45 (1.44)	3.26 (3.05)	**0.017**	**<0.001**
** Eosinophils, cells/µL**	111 (122)	183 (108)	179 (136)	0.071 ^b^	**<0.001**
** Basophils, %**	0.29 (0.17)	0.43 (0.23)	0.69 (0.22)	**<0.001 ^b^**	**<0.001**
** Basophils, cells/µL**	32 (17)	34 (16)	43 (20)	**0.034 ^b^**	0.413
** Lymphocytes, %**	14.67 (9.63)	22.5 (10.7)	32.4 (7.56)	**<0.001**	**<0.001**
** Lymphocytes, cells/µL**	1545 (801)	1689 (719)	1934 (593)	0.087 ^b^	**0.043**
** Monocytes, %**	7.17 (2.83)	8.63 (2.93)	7.89 (2.50)	0.382 ^b^	**<0.001**
** Monocytes, cells/µL**	843 (369)	720 (313)	471 (150)	**<0.0001**	**0.008**
**Platelets, ×10^9^/L**	245 (77)	228 (76)	207 (53)	0.075 ^b^	**0.001**
**N/L ratio ^#^**	6.07 [3.26–12.17]	2.66 [1.87–5.28]	1.56 [1.23–2.23]	**<0.001 ^c^**	**<0.001 ^e^**
**CRP, mg/dL**	5.82 (7.37)	11.02 (10.66)	-	-	**0.001**

All results are presented as mean (SD) unless otherwise indicated. ^#^ Median (25th–75th). ^a^ Welch’s *t* test unless otherwise specified. ^b^ Unpaired *t*-test. ^c^ Mann–Whitney test. ^d^ Paired *t* test, unless otherwise specified. ^e^ Wilcoxon test. AP, acute pancreatitis; HC, healthy control; N/L, neutrophil/lymphocyte; CRP, C reactive protein.

**Table 3 diseases-12-00018-t003:** B and T cell subsets in HCs and AP patients according to BISAP score at admission.

	BISAP^low^ (n = 34)	BISAP^hi^ (n = 16)	HC (n = 15)	*p*-Value ^a^	*p*-Value ^b^		
					BISAP^low^ vs. BISAP^hi^	BISAP^low^ vs. HC	BISAP^hi^ vs. HC
**Leucocytes, cells/µL**	11,246 (4556)	14,836 (4264)	6197 (2077)	**<0.001**	**0.013**	**<0.001**	**<0.001**
** N/L ratio ^#^**	7.09 [6.36)	11.71 (7.75)	1.89 (0.83)	**<0.001**	**0.040**	**0.018**	**<0.001**
** Neutrophils, %**	73.9 (11.86)	82.83 (9.72)	55.76 (8.23)	**<0.001**	**0.020**	**<0.001**	**<0.001**
** Neutrophils, cells/µL**	8669 (4527)	12,403 (4350)	3570 (1642)	**<0.001**	**0.009**	**<0.001**	**<0.001**
** Lymphocytes %**	16.96 (9.99)	9.81 (6.79)	32.40 (7.56)	**<0.001**	**0.025**	**<0.001**	**<0.001**
** CD4^+^ T cells, %**	60.68 (10.82)	54.14 (17.69)	64.86 (8.38)	0.058	**0.200**	0.526	0.050
** HLA-DR^+^ CD4^+^ T cells, %**	8.781 (4.47)	12.18 (6.07)	7.37 (2.93)	**0.014**	**0.048**	0.589	**0.015**
** CD45RO^−^CCR4^+^ CD4^+^ T cells, %**	3.32 (1.71)	2.05 (1.67)	3.77 (1.55)	**0.014**	**0.043**	0.683	**0.018**
** CD45RO^+^CCR4^−^ CD4+ T cells, %**	22.06 (8.88)	29.10 (11.78)	21.95 (8.54)	0.050	0.056	0.999	0.118
** Total B cells, %**	13.26 (6.66)	12.65 (5.28)	9.39 (3.65)	0.099	0.935	0.085	0.265
** CD21^dim^CD38^−^ B cells, %**	3.61 (2.25)	6.45 (4.76)	3.81 (3.48)	**0.018**	**0.017**	0.980	0.075
** CD27^−^IgD^−^ B cells, %**	2.33 (1.47)	4.09 (2.42)	2.91 (2.61)	**0.021**	**0.015**	0.629	0.240
**CRP at admission, mg/dL ^#^**	2.05 [0.38–5.50]	7.15 [1.00–16.00]	-	-	**0.024 ^c^**	-	-
**CRP at 48 h, mg/dL ^#^**	5.32 [2.15–10.23]	15.60 [7.58–28.05]	-	-	**0.010 ^c^**	-	-
**LOS, days**	4.60 (2.00)	8.00 (5.82)	-	-	**0.035 ^d^**	-	-

All results are presented as mean (SD) unless otherwise indicated. ^#^ Median (25th-75th). ^a^ One-way ANOVA. ^b^ Tukey’s multiple comparisons test. ^c^ Mann–Whitney test. ^d^ Unpaired *t*-test. AP, acute pancreatitis; HC, healthy control; N/L, neutrophil/lymphocyte; CRP, C reactive protein; LOS, length of stay at hospital; SD, standard deviation.

**Table 4 diseases-12-00018-t004:** B and T cell subsets according to the revised Atlanta classification system.

	Mild AP (n = 34)	Moderate/ Severe AP (n = 16)	HC (n = 15)	*p*-Value ^a^	*p*-Value ^b^		
					Mild AP vs. Moderate/Severe AP	Mild AP vs. HC	Moderate/ Severe AP vs. HC
**Leucocytes, cells/µL**	11,535 (4266)	14,223 (5283)	6197 (2077)	<0.0001	0.093	**<0.001**	**<0.001**
** Basophils, cells/µL**	28 (13)	41 (21)	43 (20)	0.005	**0.037**	**0.013**	0.916
**Platelets ×10^9^/L**	228 (60)	282 (95)	207 (53)	0.008	**0.029**	0.594	**0.009**
**Total CD4 T cells, %**	**60.72** **(13.12)**	54.06 (13.81)	64.86 (8.38)	0.055	0.187	0.532	**0.047**
** Total CD45RO+ CD4 T cells, %**	59.34 (12.76)	70.03 (14.04)	55.48 (11.61)	0.006	**0.022**	0.613	**0.008**
**Total B cells, %**	**12.45** **(3.40)**	14.38 (9.88)	9.39 (3.65)	0.057	0.510	0.204	**0.047**
**CRP at admission, mg/dL ^#^**	1.45 [0.29–5.53]	5.70 [2.25–10.60]	-	-	**0.035 ^c^**	-	-
**CRP at 48 h, mg/dL ^#^**	6.45 [1.83–12.00]	13.20 [4.13–29.05]	-	-	**0.035 ^c^**	-	-

All results are presented as mean (SD) unless otherwise indicated. ^#^ Median (25th–75th). ^a^ One-way ANOVA. ^b^ Tukey’s multiple comparisons test. ^c^ Mann–Whitney test. AP, acute pancreatitis; HC, healthy control; CRP, C reactive protein; SD, standard deviation.

**Table 5 diseases-12-00018-t005:** Multivariable Poisson regression analyses results: changes in the length of stay at hospital (days) in AP patients.

	Parameter	e^Estimate^	e^SD^	e^95% CI^	*p*-Value
**Model 1**	Constant	2.974	1.15	2.262–3.916	**<0.001**
	N/L ratio	1.024	1.004	1.015–1.033	**<0.001**
	B cells (%)	1.018	1.009	1.000–1.034	**0.047**
	BISAP score of 2/3	1.343	1.147	1.024–1.755	**0.031**
**Model 2**	Constant	1.734	1.192	1.224–2.440	**0.002**
	HLA-DR^+^ CD4 T cells (%)	1.034	1.012	1.011–1.057	**0.003**
	B cells (%)	1.039	1.007	1.024–1.053	**<0.001**
	BISAP [2/3]	1.459	1.141	1.126–1.890	**0.004**
	Caused by gallstones	1.349	1.152	1.021–1.780	**0.035**
**Model Performance**					
	**Pseudo-R2**	**Overdispersion**	**Residual deviance**		**AICc**
**Model 1**	0.723	0.526	26.96		208.8
**Model 2**	0.635	0.803	35.54		219.9

SD, standard deviation; AICc, Akaike information criterion.

## Data Availability

The data that support the findings of this study are available from the corresponding author upon reasonable request.

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
