# Peer review of "B Cells and Double-Negative B Cells (CD27−IgD−) Are Related to Acute Pancreatitis Severity"

_diseases, 2024, doi:10.3390/diseases12010018_

Round 1
Reviewer 1 Report
Comments and Suggestions for Authors
The manuscript by Malheiro et al. describes a potential marker of acute pancreatitis, i.e. the double-negative B cells (IgD- CD27-). The latter, tested on a larger group of patients, could become a useful tool to predict disease severity. Although tested only on 50 patients, the study is well-designed and rigorous. Statistics are clearly reported and results discussed in detail.
As a suggestion, please add a graphical abstract to attract the readers. Revise English language throughout the manuscript.
Author Response
Dear Reviewer,
Thank you very much for taking the time to review this manuscript. We have revised English language throughout the manuscript. We are uploading a Graphic Abstract as you requested.
Thank you
Reviewer 2 Report
Comments and Suggestions for Authors
The manuscript “B cells and double-negative B cells (IgD-CD27- ) are related to acute pancreatitis severity” is a research manuscript assessing novel biomarkers with a prognostic impact on acute pancreatitis. This study demonstrates that higher BISAP score, an established scoring for the clinical outcome of acute pancreatitis, is associated with increased CD27-IgD- B lymphocytes and high neutrophilic / lymphocytic ratio. Although, these findings are interesting and with a potential translational impact, there are certain issues that need to be addressed to strengthen manuscript quality.
1. The definition of BISAP in the Abstract is missing.
2. In the Introduction the authors could comment for their choice of analyzing CD27IgD B cells.
3. In the Material and Methods controls are missing
4. Stress out the novelty of the study
5. Stress out the potential translational impact of these finding in routine practice.
6. What are the limitations of the study, including the analysis both of peripheral blood and lung.

Author Response
Dear reviewer,
Thank you very much for taking the time to review our manuscript. Please find the detailed responses below and the corresponding revisions highlighted in the re-submitted files (these changes are highlitghted in yellow).
1) We have added the BISAP description in the Abstract as requested.
2) "In the Introduction the authors could comment for their choice of analyzing CD27IgD B cells".
We have added in the Introduction:
B cells have already been studied in patients with AP and the B regulatory subset of B cells has been shown to be useful in the prediction of the severity of AP[6,11]. Double‐negative B cells (CD27-IgD- B cells) are a rare B cell subset that constitutes about 5% of all peripheral B cells in healthy individuals[12].This B cell subset has been poorly characterized for a long time, until recent studies indicated their potential roles in diseases especially autoimmune diseases, some infections, and chronic inflammatory diseases[13]. As an inflammatory disease of the pancreas, it would be of interest to study this B cell subset in AP patients.
3) In Materials and Methods is mentioned:
Fifteen (n=15) age and sex-matched healthy individuals were included as the healthy control group (HC). HC were ambulatory individuals observed at Hospital da Luz Lisboa, without previous pancreatic pathology or acute systemic disease.
4) " Stress out the novelty of the study"
We have added in the Discussion:
To the best of our knowledge this is the first study to point out the changes in CD27-IgD- B cells in patients with acute pancreatitis as well as relating higher levels of this B cell subset with higher BISAP scores.
5) "Stress out the potential translational impact of these finding in routine practice".
We have also addded in the Discussion:
As CD27-IgD- B cells may be involved in various diseases, including chronic inflammatory and infectious diseases, autoimmune diseases as well as some neoplasms, research on this subset of B cells might be essential in understanding these conditions and their treatments. As an example, therapies targeting CD27-IgD- B cells may improve clinical outcomes in chronic conditions, while understanding the reasons why this subset of B cells is increased locally in some cancers may provide insights into these neoplasms progression and eventually develop new immunotherapeutic strategies.
6) "What are the limitations of the study, including the analysis both of peripheral blood and lung".
In the Discussion in the paragrah where we analyze the main limitations of our study we have added:
Unlike neoplastic diseases, the acquisition of tissue samples from patients with acute pancreatitis is difficult and not feasible, which limits scientific research to some extent. Therefore, evidence of immunological alternations in patients with acute pancreatitis has been made mainly by studying peripheral blood. Peripheral blood, besides being accessible to study, also reflets the systemic component of acute pancreatitis as previously described.
We hope to have responded to the changes kindly requested to the manuscript.
Kind regards
Reviewer 3 Report
Comments and Suggestions for Authors
This is an interesting and well conducted study regarding alterations in the immune profile, namely in the circulating B cell compartment of AP patients within the first 48 hours of diagnosis. The authors constructed a model for prediction of hospital stay according to which a higher BISAP score, the N/L ratio and the frequency of peripheral blood B cells were the best predictors of length of stay of AP patients.
Overall, the manuscript is well written. The materials and methods are well described and the results are clearly presented. The discussion is comprehensive and the conclusions are based on the findings of the study.
Detailed Comments:
1. In this study, the authors evaluated the possible implication of peripheral B and T cell subsets in acute pancreatitis (AP) by measuring their populations by flow cytometry. Fifty AP patients were included; measurements were made on admission and after 48 hours and compared with 15 healthy controls.
On admission, AP patients showed decreased percentages of CD4+ T cells and increased percentages of B cells compared to control subjects. After 48 hours there was an increase in T cell and CD4+ T cell percentages, and a decrease in the percentages of activated HLA-DR+CD4+ T cells compared with the values on admission.
Differential variations in the cellular immune distributions were found according to the severity of AP as measured by the Bedside Index of Severity in Acute Pancreatitis (BISAP) score.
Multivariable regression analysis was used to evaluate the association between the immune profile and the length of hospital stay. The constructed model showed that higher BISAP score, the N/L ratio and the frequency of peripheral blood B cells were the best predictors of length of stay of AP patients.
2. Manuscript’s strengths and weaknesses.
Overall, the manuscript is well written and the subject is interesting. The materials and methods are well described and the results are clearly presented. The discussion is comprehensive and the conclusions are based on the findings of the study. The discussion is comprehensive even if extended to some degree and the conclusions are based on the results.
The main weakness of the study is the relatively small number of patients and controls. Apparently larger numbers are necessary to provide a solid evidence that these parameters are useful markers for AP severity and length of hospital stay.
3. Point-by-point recommendations for the improvement of the manuscript
In the Abstract “BISAP” should be given n full.
Proper use of “dot” in numbers instead of “comma” should be used (see tables)
The caption in Figure 4 should be more descriptive.
Author Response
Dear Reviewer,
Thank you very much for taking the time to review this manuscript.
Kind regards
Reviewer 4 Report
Comments and Suggestions for Authors
Acute pancreatitis (AP) is a serious and frequent disease, which generates high costs in the health system. Currently, there is a lack of proper prediction markers for AP patients.
This research aimed to explore peripheral immune cell subsets in AP patients and assess their association with disease severity.
In detail, the goal of the study was to evaluate whether immune cells can be used to determine severity and length of hospital stay early in the course of the disease and therefore be used as biomarkers of acute pancreatitis severity. In this prospective observational study based on clinical results the authors described the importance of B and T cell subsets as the potential determinants of severity and length of hospital stay early in the course of the AP disease. The authors developed a model according to a higher BISAP (Bedside Index of Severity in Acute Pancreatitis) score, the N/L (neutrophil/lymphocyte) ratio and the frequency of peripheral blood B cells.
The study is interesting and can be the foundation for further analyses based on bigger patent cohort. The study is well-designed and written. Introduction and methods are clearly prepared. The discussion is a little bit long, but it correctly presents the study and related references.
The number of patent included in the research (50 AP + 15 control) is to small to definitely state the usefulness of parameters as the markers for AP patients, but it is a good pilot study. The authors discussed this limitation properly. I think that abstract and small issues concerning tables and figures should be corrected (see Minor issues). I have no major issues to report.
Minor issues:
1) In abstract: BISAP abbreviation should be described when first time mentioned; abstract should be without headings, like Background, Methods...; acstract should be shortened to the word limit 200.
2) In Table 1 please correct Alchoolic to Alcoholic.
3) Often the authors used comma (,) instead of dot (.) in numbers, especially in tables; please, change to dot.
4) In Figure 4 - description should define A and B part of Figure.
Author Response
Dear reviewer,
Thank you very much for taking the time to review this manuscript. Please find the detailed revisions below:
1) We have described the BISAP as requested as well as have rewriten the Abstract without Headings and shorted to 200 words.
Abstract: Acute pancreatitis (AP) is an increasingly frequent disease in which inflammation plays a crucial role. Fifty patients hospitalized with AP were included and peripheral blood samples were analyzed for B and T cell subpopulations at time of hospitalization and 48 hours after diagnosis. The Bedside Index of Severity in Acute Pancreatitis (BISAP) and length of hospital stay were also recorded. A healthy control group (HC) of 15 outpatients was included. AP patients showed higher neutrophil/lymphocyte (N/L) ratios and higher percentages of B cells than HC. Total B cell percentages were higher in patients with moderate/severe AP than in patients with mild AP. The percentages of B cells decreased from admission to 48 hours after admission as well as the percentages of the CD27-IgD- B cell subset. Patients with higher BISAP scores showed lower percentages of peripheral lymphocytes but higher percentages of CD27-IgD- B cells. Higher BISAP scores, N/L ratio and peripheral blood B cell levels emerged as predictors of hospital stay length in AP patients. Our findings underscore the importance of early markers for disease severity. Addicionally, the N/L ratio along with the BISAP score and circulating B cells, form a robust predictive model for hospital stay duration in AP.
2), 3) and 4) we have made the changes as you kindly requested and they can be found in the revised manuscript.
Thank you
Kind regards